# Potentials of Natural Antioxidants in Reducing Inflammation and Oxidative Stress in Chronic Kidney Disease

**DOI:** 10.3390/antiox13060751

**Published:** 2024-06-20

**Authors:** On Ying Angela Lee, Alex Ngai Nick Wong, Ching Yan Ho, Ka Wai Tse, Angela Zaneta Chan, George Pak-Heng Leung, Yiu Wa Kwan, Martin Ho Yin Yeung

**Affiliations:** 1Department of Health Technology and Informatics, The Hong Kong Polytechnic University, Hong Kong SAR, China; on-ying-angela.lee@connect.polyu.hk (O.Y.A.L.);; 2Department of Anatomical and Cellular Pathology, Prince of Wales Hospital, The Chinese University of Hong Kong, Hong Kong SAR, China; 3Department of Pharmacology and Pharmacy, The University of Hong Kong, Hong Kong SAR, China; gphleung@hku.hk; 4The School of Biomedical Sciences, The Chinese University of Hong Kong, Hong Kong SAR, China

**Keywords:** chronic kidney disease, antioxidants, inflammation, oxidative stress, renal preservation

## Abstract

Chronic kidney disease (CKD) presents a substantial global public health challenge, with high morbidity and mortality. CKD patients often experience dyslipidaemia and poor glycaemic control, further exacerbating inflammation and oxidative stress in the kidney. If left untreated, these metabolic symptoms can progress to end-stage renal disease, necessitating long-term dialysis or kidney transplantation. Alleviating inflammation responses has become the standard approach in CKD management. Medications such as statins, metformin, and GLP-1 agonists, initially developed for treating metabolic dysregulation, demonstrate promising renal therapeutic benefits. The rising popularity of herbal remedies and supplements, perceived as natural antioxidants, has spurred investigations into their potential efficacy. Notably, lactoferrin, *Boerhaavia diffusa*, *Amauroderma rugosum*, and *Ganoderma lucidum* are known for their anti-inflammatory and antioxidant properties and may support kidney function preservation. However, the mechanisms underlying the effectiveness of Western medications and herbal remedies in alleviating inflammation and oxidative stress occurring in renal dysfunction are not completely known. This review aims to provide a comprehensive overview of CKD treatment strategies and renal function preservation and critically discusses the existing literature’s limitations whilst offering insight into the potential antioxidant effects of these interventions. This could provide a useful guide for future clinical trials and facilitate the development of effective treatment strategies for kidney functions.

## 1. Introduction

Chronic kidney disease (CKD) poses a significant global public health challenge, associated with high morbidity and mortality [1]. It is characterized by impaired renal function, with signs of kidney damage persisting for more than 3 months, as indicated by albuminuria (albumin excretion ≥ 30 mg/g) and/or reduced estimated glomerular filtration rate (eGFR) (eGFR < 60 mL/min/1.73 m^2^) [2,3]. If left untreated, CKD can progress to end-stage renal disease (ESRD), necessitating long-term dialysis or kidney transplantation.

CKD can result from various conditions, including hypertension, diabetes, polycystic kidney diseases, and glomerulonephritis [4]. These conditions contribute to the rising caseloads of CKD and impose substantial healthcare expenditures. The Global Burden of Disease Study 2017 reported a 33% increase in the worldwide prevalence of CKD between 1990 and 2017 [5].

The progression of CKD is often associated with inflammation and oxidative stress, which contribute to the worsening of kidney function. This can lead to a heightened risk of cardiovascular disease, neuropathy, retinopathy, hypertension, anaemia, bone disease, and susceptibility to infections [6,7,8,9]. In severe cases, patients may require kidney transplantation or lifelong dialysis. These complications can significantly impact patients’ quality of life, with high morbidity and mortality rates [10]. Furthermore, CKD imposes a considerable financial and medical resource burden on both society and individual patients, with treatment costs varying widely depending on disease severity [11].

The progression of CKD is heavily influenced by inflammation and oxidative stress [12]. Inflammation is the immune system’s defensive response to tissue damage or infection, whilst oxidative stress occurs due to an imbalance between reactive oxygen species (ROS) generation and the body’s antioxidant mechanisms. Hyperglycemia is one of the most common risk factors for having CKD [13]. For patients with hyperglycemia, the elevated blood glucose levels activate inflammatory pathways that stimulate the production of pro-inflammatory cytokines, adipokines, and chemokines, which attract immune cells to the kidneys [14]. Neutrophils and monocytes generate reactive intermediates that contribute to the release of ROS, resulting in oxidative stress and subsequent kidney damage [15]. Additionally, elevated glucose levels activate the polyol pathway, causing the formation of sorbitol, which intensifies oxidative stress and inflammation in the kidneys [16]. By reducing inflammation and oxidative stress, it may be possible to mitigate the development and progression of CKD. Glycosylation in the kidney resulting from the overactivated hexosamine pathway caused by an increased UDP-GlcNAc production in the renal system can lead to the production of advanced glycation end-products (AGEs) [17], which triggers inflammation, oxidative stress, and fibrosis that negatively impact renal function and health [18]. The accumulation of AGEs due to hexosamine pathway dysregulation in the kidneys can thus promote CKD progression through pathogenic mechanisms like renal inflammation and fibrosis [19]. These pathways can also activate signalling pathways, including the nuclear factor-kappa (NF-κB) and transforming growth factor-β (TGF-β1) pathways that are associated with kidney damage [20].

Despite conventional antihypertensive therapies such as angiotensin-converting enzyme (ACE) inhibitors, angiotensin II receptor blockers (ARBs), and glucose-lowering drugs like metformin, a considerable number of patients unfortunately still progress to ESRD [21]. Therefore, there is an urgent need for novel therapeutic approaches to enhance CKD treatment outcomes by more directly targeting these pathological processes. Antioxidants have garnered significant interest as potential therapeutic agents for CKD due to their ability to counteract oxidative stress and inflammation. Numerous studies have explored the efficacy of antioxidants in preventing or delaying the progression of CKD by targeting oxidative stress and inflammation, and they show promising results. However, further research is necessary to identify the optimal antioxidant agents and dosages for CKD treatment and to elucidate the underlying mechanisms of their beneficial effects. The electronic databases PubMed (via MEDLINE) and Web of Science (WOS) were searched to identify relevant literature published up until May 2024. In PubMed, medical subject heading (MeSH) terms were used to capture articles related to various antioxidant compounds and chronic kidney disease treatments. The MeSH terms searched included “Antioxidants”, “Chronic Kidney Disease”, “Lactoferrin”, “*Boerhaavia diffusa*”, “*Amauroderma rugosum*”, “*Ganoderma lucidum*”, “Statins”, “Metformin”, “Glucagon-like peptide 1 Agonists”, “ACE inhibitors”, “ARBs”, and “Sodium-glucose co-transporter 2 Inhibitors”. In WOS, the terms “Antioxidants”, “Chronic Kidney Disease”, and “Reactive Oxygen Species” were searched.

## 2. Oxidative Stress and Inflammation in CKD

CKD is a growing health problem characterized by the progressive loss of renal function over time [22,23]. Oxidative stress is a crucial factor in the onset and progression of CKD. It occurs when there is an imbalance between the production of ROS and the body’s ability to neutralise them [24]. Under normal physiological conditions, ROS are formed as natural byproducts of oxygen metabolism in the mitochondria and various cellular processes. At low levels, ROS serve important functions in cell signalling, gene expression, cell differentiation, apoptosis, and muscle power regulation [25,26,27].

ROS include free radicals like superoxide (O2^•−^), hydroxyl radical (HO^•^), hydrogen peroxide (H_2_O_2_), singlet oxygen (^1^O_2_), and nitrogen oxide (NO^•^) [27,28]. They are generated endogenously through cellular processes in organelles like mitochondria; enzymes in peroxisomes such as acyl CoA oxidases and urate oxidase [29]; and enzymes in the endoplasmic reticulum such as cytochrome p-450, b5 enzymes, and diamine oxidase [30]. Exogenous ROS can also originate from environmental sources such as contaminants, radiation, diet, and medications [31].

The human body has a tightly regulated antioxidant system neutralising ROS and maintaining them at non-toxic levels. However, in conditions of chronic disease, mitochondrial dysfunction, and exposure to exogenous ROS, excess ROS generation overwhelms antioxidant capacity. This imbalance leads to oxidative damage to biological molecules like lipids, proteins, and DNA [32]. Given the kidneys’ high metabolic activity and numerous oxidation processes in mitochondria, they are particularly vulnerable to oxidative stress [28,33,34]. Excess ROS can also contribute to CKD via three major pathological pathways, inflammatory, fibrotic, and apoptotic, and will be discussed below [35] (Figure 1).

### 2.1. Inflammatory Pathways

Excessive ROS trigger the activation of TGF-β from its latent to its phosphorylated form [37]. Activated TGF-β then promotes the formation of the NLRP3 inflammasome, a multi-protein complex in the innate immune system capable of activating inflammatory caspases such as caspase-1 and various cytokines [38]. Upon activation, NLRP3 cleaves and autoproteolysis caspase-1 fragments from pro-caspase-1 to active caspase-1 [38]. Active caspase-1, in turn, cleaves the pro-domains from pro-IL-1β and pro-IL-18, generating mature active forms of pro-inflammatory cytokines via proteolysis (IL-1β and IL-18) that play important roles in increasing inflammation and immunological responses in CKD [39,40,41,42]. The release of tumour necrosis factor-alpha (TNF-α) is stimulated by IL-1β and IL-18 from immune cells, further contributing to inflammation. TNF-α is a major proinflammatory cytokine elevated in CKD [43], perpetuating the inflammatory response in kidneys by inducing the expression of chemokines, adhesion molecules, and cytokines such as IL-1β and IL-18. This exacerbates CKD by amplifying the inflammatory response and immune cell infiltration.

Both IL-1β and IL-18 play a significant role in polarizing T-helper cell responses [44]. IL-1β non-specifically amplifies both T and B lymphocytes, whereas IL-18 substantially activates T helper 1 immune response via IFNγ production [45,46,47]. The IFNγ produced by IL-18 would serve as a chemoattractant, attracting immune cells like macrophages to the site of inflammation [48]. Moreover, IFNγ may upregulate the expression of adhesion molecules (ICAM-1, VCAM-1, and E-selectin) on endothelial cells lining the kidney’s blood vessels, facilitating immune cell adherence and migration to the kidneys [49]. Synergistically with TNF-α and IL-1β, IFNγ further enhances the inflammatory response and exacerbates direct cellular damage to the kidneys. In CKD, immune cells like monocytes, macrophages, neutrophils, NK cells and Th1/17 cells infiltrate to the kidneys, causing damage by releasing cytokines, ROS, autoantibodies, and serine proteases (such as granzymes) [50,51]. The production of these cytokines and chemokines attracts immune cell infiltration into kidney tissues, including macrophages and T cells, further contributing to the development of CKD through prolonged inflammation and tissue damage.

### 2.2. Fibrotic Pathway

Fibrosis plays a pivotal role in the development and progression of CKD. The pathological accumulation of extracellular matrix proteins in the kidney leads to functional and structural impairment. There are two major signalling cascades mediating fibrosis: the canonical TGF-β/Smad pathway and non-canonical pathways like JNK and NF-κB. The sustained hyperactivation of these pathways drives exacerbated matrix deposition through different mechanisms, eventually resulting in CKD. The pathways are further discussed below.

#### 2.2.1. Canonical (Classical) Smad Pathway

The canonical TGF-β signalling pathway, known as the Smad-dependent pathway, primarily drives TGF-β-mediated fibrosis [52]. Excessive ROS activate TGF-β from its latent to its active phosphorylated form [53]. Active TGF-β signals through TGF-β type I and II serine/threonine kinase receptors on target renal cells. This activation phosphorylates and activates Smad2/3 proteins, forming complexes with Smad4 and translocating to the nucleus.

Within the nucleus, these complexes upregulate the transcription of profibrotic genes such as CTGF, COLI, and αSMA [54,55]. COLI encodes for type I collagen, the most abundant protein in fibrous tissues. Under TGF-β1 stimulation, excessive COLI expression leads to its overproduction and accumulation in the extracellular matrix of blood vessels, glomeruli, and kidney interstitium, resulting in renal fibrosis [56,57]. αSMA is the isoform of actin that is expressed strongly in smooth muscle cells, myofibroblasts, and activated fibrogenic cells [58,59]. The upregulation of αSMA marks the transformation of resident fibroblasts into collagen-producing myofibroblasts, driving fibroblasts. CTGF promotes collagen synthesis, fibrotic lesion maintenance, and myofibroblast differentiation downstream of TGF-β1 signalling [59]. Increased CTGF exacerbates fibrosis [60,61] and promotes extracellular matrix deposition and fibrosis in the kidneys.

#### 2.2.2. Non-Canonical (Non-Classical) JNK and NF-κB Pathway

In addition to the canonical Smad pathway, excess ROS can stimulate non-canonical pathways to drive fibrotic gene expression [62]. The major non-canonical pathways involved are the JNK pathway and the NF-κB pathway. Both pathways are stimulated independently of Smad through distinct mechanisms of ROS generation.

#### 2.2.3. JNK Pathway

The JNK signalling pathway is stimulated downstream of TGF-β. Upon activation, JNK phosphorylates transcription factors like c-Jun and c-Fos, forming the AP-1 complex [63]. This complex increases the release of pro-inflammatory factors such as TNF-α and CCL2 and pro-fibrotic factors such as TGF-β and CTGF [64]. The persistently active JNK signalling pathway stimulates the increased production of chemokines and cytokines that cause inflammation and fibrosis in the kidneys [63].

#### 2.2.4. NF-κB Pathway

ROS can directly activate NF-κB signalling by inducing phosphorylation and degradation of its endogenous inhibitor IκB [65]. This allows NF-κB to translocate from the cytosol to the nucleus and function as a transcription factor. NF-κB is activated in response to different stimuli, including elevated ROS levels during oxidative stress [66,67]. Activated NF-κB perpetuates renal inflammation and fibrosis by upregulating NF-κB and pro-inflammatory mediators (TNF-α) and inflammatory cytokines (IL-1β and IL-18) [68,69]. These cytokines can induce the expression of chemokines and adhesion molecules in kidney tissues, boosting immune cell recruitment and aggravating chronic inflammation. Moreover, they would also promote renal interstitial fibrosis and elevate proinflammatory chemokines and cytokines via increasing CTGF expression [70].

By perpetuating renal inflammation and fibrosis, persistent NF-κB activation mediated by ROS or JNK signalling exerts profibrotic effects and contributes to the pathogenesis and progression of CKD over time. NF-κB functions as a key modulator of the oxidative stress-induced fibro-inflammatory response.

The cumulative effects of TGF-β driven Smad and non-Smad promote excessive accumulation of extracellular matrix proteins that damage normal renal structure over time and eventually lead to CKD.

### 2.3. Apoptotic Pathway

Apoptosis is a tightly regulated programmed cell death process that can be activated through intrinsic or extrinsic pathways [71]. Under oxidative stress, both routes are stimulated, leading to renal cell loss and CKD progression.

#### 2.3.1. Intrinsic Pathway

The intrinsic pathway involves the release of pro-apoptotic proteins from mitochondria in response to ROS generation. This triggers the caspase cascade within the cell and leads to renal cell death. ROS activate the pro-apoptotic protein BIM, which in turn activates the pore-forming proteins BAK and BAX in the mitochondria, inducing mitochondrial outer membrane permeabilization [71,72]. This triggers cytochrome c release from mitochondria into the cytosol, where it binds to Apaf-1 [73]. The interaction between cytochrome c and Apaf-1 leads to a conformational shift in Apaf-1, allowing it to oligomerize into a heptameric structure and recruit procaspase-9. Together, this forms the apoptosome complex [74].

Overexpression of BAX and BAK results in mitochondrial outer membrane permeabilization and cytochrome c release, further promoting apoptosome formation [75]. By dimerizing in the apoptosome complex, procaspase-9 is recruited and activated through phosphorylation into caspase-9 [76].

Apoptosome activation through caspase-9 phosphorylation leads to its cleavage and activation by the apoptosome complex. Activated caspase-9 then propagates the caspase cascade by proteolytically cleaving and activating downstream executioner caspases like caspase-7, -6, and -3 [77,78]. These executioner caspases execute the apoptotic program through proteolytic degradation of critical cellular proteins. Specifically, they dismantle DNA repair enzymes, structural scaffolding proteins, and activate DNases, leading to the fragmentation of nuclear material [79,80,81]. Together, this caspase activation ultimately results in programmed cell death [62,82]. The apoptotic cell loss of tubular epithelial cells and endothelial cells contributes to renal failure over the extended periods seen in CKD.

Through apoptosome formation, ROS are thus able to trigger an intrinsic apoptotic signalling cascade, culminating in caspase activation and cellular apoptosis, promoting progressive renal deterioration in CKD.

#### 2.3.2. Extrinsic Pathway

ROS can indirectly activate the extrinsic apoptosis pathway in multiple ways. First, ROS promotes ubiquitin-mediated degradation of the caspase-8/-10 inhibitory protein c-FLIP [83]. With the downregulation of c-FLIP, the adaptor protein FADD can be more readily bound to pro-caspase-8/-10 at the death-inducing signalling complex (DISC) upon death receptor ligation, enhancing caspase-8/-10 recruitment and activation [71,83]. ROS also stimulates the release of the death ligand TNF-α by activating NF-kB-mediated inflammation [84]. TNF-α then binds to its receptor TNF-R1 of renal cells.

In the extrinsic pathway, death ligands like FasL, TNF-α, and TRAIL bind to their cognate cell surface death receptors FasR, TNF-R1, and DR4. Upon ligand binding, dimerization of these death receptors occurs, allowing recruitment and activation of initiator caspase-8/-10, which subsequently activates the downstream caspase-3, propagating the apoptotic cascade [78,85,86,87].

Like the intrinsic pathway, caspase-3 activation is the convergent point between both apoptotic routes [78,85]. Execution of the cell death program by caspase-3 leads to renal cell loss, contributing to the progressive renal deterioration seen in CKD over time. The extrinsic apoptotic pathway causes tubular atrophy and endothelial injury, which subsequently results in CKD development [88,89,90].

In summary, oxidative stress significantly contributes to CKD development and progression through multiple interconnected pathways. Excess ROS activate the inflammatory pathways involving cytokines such as IL-1β, IL-18, IFNγ, and TNFα. These cytokines promote leukocyte infiltration, perpetuate inflammation, and cause direct cytotoxicity. ROS also stimulate the profibrotic signalling cascades mediated by TGF-β and NF-kB, resulting in renal fibrosis. Additionally, both intrinsic and extrinsic apoptotic pathways are activated by oxidative stress, leading to tubular epithelial cell death. These mechanisms converge to elicit glomerular damage, tubulointerstitial fibrosis, loss of kidney function, and ultimately CKD development.

## 3. Metabolic Therapeutics: Unveiling Mechanisms for CKD Management

In the pathophysiology of CKD development, the role of inflammation is increasingly recognized as a fundamental driver in disease progression. Originating from conditions associated with metabolic syndrome such as obesity, hypertension, and diabetes, inflammation can lead to renal damage, resulting in suboptimal outcomes for CKD patients. To address these issues, several therapeutic agents initially designed to target metabolic syndrome abnormalities, including statins, metformin, glucagon-like peptide 1 (GLP-1) agonists, ACE inhibitors (ACEi), ARBs, and sodium-glucose co-transporter 2 (SGLT2) inhibitors, have been integrated into clinical treatments. Beyond their primary metabolic efficacies, these drugs exhibit promising benefits for renal health, notably due to their anti-inflammatory and antioxidant properties. This section delves into the biochemical mechanisms and clinical data surrounding these therapeutic agents, highlighting their emerging significance in CKD therapeutics. To offer a comprehensive overview of emerging therapeutic agents in CKD treatment, a table summarizing current findings on various kidney diseases, their related metabolic targets/pathways involving oxidant stress, and their functional roles in alleviating CKD is included (Table 1).

### 3.1. Statin

Statins, primarily known as hydroxymethylglutaryl-CoA (HMG-CoA) reductase inhibitors, are important in the regulation of cholesterol biosynthesis by inhibiting the rate-limiting enzyme—HMG-CoA reductase [91]. This inhibition leads to reduced cholesterol biosynthesis, subsequently amplifying the expression of low-density lipoprotein (LDL) receptors in hepatocytes, thereby facilitating LDL cholesterol clearance from the bloodstream [92].

Recently, systematic reviews and meta-analyses have illustrated the beneficial role of statins in enhancing renal functions through cholesterol reduction mechanisms and non-cholesterol-mediated mechanisms. A vast number of research works have proved the effectiveness of statins in reducing CKD progression, cardiovascular disease development risk, and mortality [93,94,95]. The renoprotective ability of statins was mostly reflected by a reduction in albuminuria and enhanced glomerular filtration rate in various renal disease models. Among the statin family, members like atorvastatin, rosuvastatin, simvastatin, and cerivastatin are found to alleviate different renal diseases, and they differ in potency, side effects, and drug interaction tendencies. Nevertheless, while the benefits of statins on early-stage CKD are acknowledged, the effect of statins on end-stage kidney failure is still unclear [96].

In the cholesterol reduction mechanism, due to the lipid-lowering ability of statins, they are clinically effective drugs for CKD associated with hyperlipidaemia and dyslipidaemia. As an inhibitor for the essential enzyme, HMG-CoA, involved in cholesterol synthesis, statins attenuate the production of endogenous cholesterol. Hence, statins effectively regulate plasma cholesterol levels, demonstrated by lowered LDL-C, lowered triglycerides, and raised HDL-C, in CKD patients, regardless of any dialysis treatment received [93,97]. Through relieving hypercholesterolaemia or dyslipidaemia conditions in CKD patients, statins ameliorate major atherosclerotic events [98,99]. Hence, they are effective in restoring renal blood flow, which enhances glomerular filtration and improves renal deformities such as inflammation and tubular defects. The boost in renal function is also reflected by reduced proteinuria and increased eGFR, which contributes to reducing the mortality and morbidity rate of CKD patients.

On top of reducing cholesterol levels, the inhibition of HMG-CoA by statins also suppresses downstream synthesis of isoprenoids, a protein involved in intracellular signalling for a variety of gene expressions [97]. This relates to the non-cholesterol-mediated renoprotective mechanisms of statins, referring to the alleviation of CKD progression independent of cholesterol reduction due to the antioxidant effect of statins. Statins inhibit the activities of pro-oxidant enzymes, such as NADPH oxidases, and pro-inflammatory chemokines. In addition, they upregulate the expression of antioxidant enzymes like the NRF2 signalling pathway, resulting in raised levels of catalase and SOD [100,101]. Combining statin’s action towards prooxidant and antioxidants, it reduces ROS accumulation and alleviates cellular senescence in renal disease progression [100]. For example, it is found that Atorvastatin alleviated diabetic kidney disease (DKD) and diabetic nephropathies through downregulating pro-oxidant expression, including histone deacetylase (HDAC), NADPH oxidases, and Nox4, along with promoting cellular survival or proliferation by activation of the Akt/GSK3β and RhoA signalling pathway, resulting in elevated renal e-cadherin expression that indicated restored kidney function [102,103,104]. Rosuvastatin is also suggested to effectively treat diabetic nephropathy by lowering 8-OhdG levels, indicating decreased oxidative stress accumulation [105]. Consequently, reduced albuminuria and glomerular hypertrophy accompanied by improved insulin resistance are observed after drug treatment.

Moreover, statins inhibit cellular apoptosis by regulating apoptosis-associated pathways. During the disease course of CKD, apoptosis of renal cells is promoted, leading to cellular death that is associated with tubular atrophy and renal fibrosis. Cormack-Aboud and his team revealed the effect of rosuvastatin on the slower progression of glomerulosclerosis in adriamycin- and puromycin aminonucleoside-induced CKD, mainly by the p21-dependent antiapoptotic pathway that exerted anti-apoptotic and pro-survival effects on podocytes [106]. Treatment with Simvastatin helped reduce apoptosis in renal endothelial progenitor cells, contributing to reversing endothelial dysfunction and renal damage [107]. Atorvastatin also reduces apoptosis by suppressing the ERK1/2 pathway that prevents renal parenchymal cell loss [102].

In addition, statins exert anti-inflammatory effects on the kidneys. Statins suppress the signalling pathways involved in the production of pro-inflammatory cytokines and mediation of immune cell infiltration. This is exemplified by the contribution of simvastatin to the downregulation of both COX 2 and TNF-α expression, thereby reducing the production of pro-inflammatory mediators such as prostaglandins [108]. This decelerated CKD progression is caused by the effect of angiotensin II in human mesangial cells. Furthermore, other than the ROS-reduction activity, Atorvastatin diminished sickle cell nephropathy by declining Cybb levels, which is associated with activated phagocytes [109]. Cerivastatin also participated in the alleviation of diabetic nephropathy with hypertension by acting against inflammatory pathways such as the MCP-1- and TGF-β-related signalling pathways [110]. It was found that Cerivastatin restored the charge barrier of GBM, thus contributing to the prevention of glomerular hyperfiltration and various renal histological abnormalities such as mesangial expansion.

### 3.2. Metformin

Metformin is a first-line treatment for diabetes, primarily reducing blood sugar levels [111] by suppressing gluconeogenesis and decreasing hepatic glucose output. Its mechanism involves entry into hepatocytes via the OTC1 transporter [112] and subsequent accumulation, especially within mitochondria [113]. Once inside, metformin inhibits the mitochondrial respiratory complex I, altering the ADP to ATP ratio and activating AMPK, which in turn inhibits gluconeogenesis [114].

Metformin’s potential in renal protection has garnered significant clinical attention. It was found that metformin reduced all-cause mortality risk and cardiovascular events in CKD, especially in DKD of diabetic stage G3 [115,116,117]. The effect of metformin on CKD is also reflected by decelerated renal dysfunction progression and attenuated eGFR decline upon drug treatment [118,119].

Recent research has made promising steps in deciphering the mechanistic basis by which metformin potentially exerts renoprotection, mainly attributing to its direct antioxidative activity and indirect antioxidative activity by the modulation of the lipid or glucose metabolism. Acting as an antioxidant, metformin exhibited excellent ability in the suppression of ROS accumulation in drug-induced renal injury, including folic acid- and adenine-induced CKD and DKD. ROS accumulation in the mitochondria was especially alleviated by metformin as the drug inhibited complex I-dependent respiration [120,121]. As mitochondrial ROS generation is reduced, mitochondrial-mediated apoptosis is prevented [122]. Various research has demonstrated the effectiveness of metformin in CKD treatment. For instance, it mitigated mitochondrial oxidative stress by reducing the activity of N-acetyl-β-D-glycosaminidase and preventing the depletion of cytochrome c and NADH, resulting in an enhanced mitochondrial balance in gentamicin-induced renal injury [123]. Such a reverse in renal injury was reflected by raised eGFR and renal blood flow.

Moreover, it regulates inflammatory pathways and exerts its anti-inflammatory activity in CKD alleviation. Studies have suggested the role of metformin in the diminishing activation of immune cells and the release of pro-inflammatory cytokines through mechanisms such as inhibition of P38 phosphorylation and NF-κB, reflected by the decline of MCP-1, F4/80, and ICAM1 [124,125,126]. Another study conducted by Zhou further elaborated on the action of metformin to inhibit NF-κB through an AMPK-dependent pathway in DKD [126].

Furthermore, the anti-fibrotic action of metformin on renal injuries has been widely investigated. It was found that it relieved the overexpression of TGF-β through Smad and non-Smad mechanisms, which reduced collagen production induced by TGF-β, thereby limiting the development of renal interstitial fibrosis and other associated pathologies [124,127,128]. The anti-fibrotic activity of metformin was also indicated through decreased fibrotic indicators such as type IV collagen, fibronectin, and CTGF in the kidney in other studies [124,129,130]. With renal histology and renal function markers being evaluated in these studies, metformin exhibited significant improvement in renal histology and parameters such as serum creatinine, urinary creatinine, and blood urea nitrogen.

On top of metformin’s antioxidant activity, it modulates lipid metabolism, which demonstrates its significance in fat-induced or diabetic renal diseases. It is revealed that metformin countered lipotoxicity through concurrent activation of AMPK/PPARα routes and inhibition of the SREBP1 and FAS pathways, as well as enhancing mesangial GLP-1R expression, which reduces mesangial cell apoptosis [131,132]. Also, it changed glycolipid metabolism by upregulating AMPK and SIRT1 expression and downregulating FoxO1 expression [124,133]. Subsequently, renal histological improvements such as mesangial expansion, tubular dilations, and interstitial fibrosis were ameliorated, accompanied by a decline in apoptotic and fibrotic markers. Alteration in these pathways crucial for lipid metabolism control demonstrated the role of metformin in tackling the underlying cause of CKD, thereby moderating the development of downstream pathological activities such as ROS accumulation and abnormal cell proliferation.

### 3.3. Glucagon-like Peptide 1 Agonists

GLP-1 receptor agonists are potent therapeutics commonly used in managing T2DM and certain cases of obesity. Engineered to mimic the gut-derived peptide hormone GLP-1, these agonists enhance insulin secretion from pancreatic islets in response to oral glucose intake. This action, known as the incretin effect, helps lower blood glucose levels. GLP-1 agonists are divided into two categories based on their structure. One category consists of agents derived from the human GLP-1 backbone and include Dulaglutide, Albiglutide, Liraglutide, and Semaglutide. Another category contains those rooted in the exendin-4 backbone, comprising Exenatide (in two formulations) and Lixisenatide [134]. Synthetic GLP-1 agonists are designed to resist degradation by the dipeptidyl peptidase 4 (DPP-4) enzyme, thereby having a longer half-life than the natural peptide hormone [135]. These agonists work by interacting with specific GLP-1 receptors on target tissues such as the pancreas. There, they stimulate insulin biosynthesis and secretion from beta cells and suppress glucagon secretion from alpha cells. In addition, GLP-1 agonists act on other tissues. In the liver, they limit hepatic gluconeogenesis, and in the brain they provide neuroprotective effects [136]. Through these various mechanisms, GLP-1 agonists offer effective solutions for metabolic regulation in conditions like type 2 diabetes mellitus and obesity.

Recent meta-analyses further provide growing evidence supporting the renal benefits of GLP-1 agonists against CKD associated with T2DM. GLP-1 agonists show clinical significance in treating T2DM-associated CKD as they contribute to a reduction in all-cause mortality and provide cardiovascular benefits to CKD patients [137]. Improved renal injuries were also observed through a reduction in albuminuria and enhanced eGFR [137,138]. It also prevented the deterioration of CKD by reducing the risk of macroalbuminuria development and ESRD progression without exposing patients to increased risks of severe hypoglycaemia, pancreatic adversities, or thyroid cancer [138,139].

GLP-1 contributes to slowed DKD progression through direct antioxidant activity. Due to its inhibitory effect on protein kinase C (PKC)-β and concurrent amplificatory action on protein kinase A (PKA), Beinaglutide reduced NADPH oxidases activation, leading to a diminished ROS and glycation end-product accumulation in renal glomeruli and tubules [140,141]. The antioxidant effect of GLP-1 is also supported by the enhancement in oxidative stress metabolites, such as NAD+ and adenosine, and cardiolipin essential for mitochondrial metabolism after Semaglutide and Dulaglutide treatment, respectively [142,143]. Reduction in ROS correlated with the lowered urinary albumin/creatinine ratio and alleviated progression in glomerulopathy. Moreover, it is revealed that liraglutide normalized Nox4 levels. In addition to the strengthened antioxidant defence, this favours vasodilation, further contributing to the alleviation of hypertension, inflammation, and fibrosis [144].

Studies also revealed GLP-1 agonists’ anti-inflammatory mechanism of CKD prevention. Diabetic nephropathy was alleviated by GLP-1 via inhibition of inflammatory activities such as MAPK and NF-κB phosphorylation, contributing to anti-inflammatory effects on podocytes and tubular cells, thereby reducing glomerulosclerosis and renal tubulointerstitial injuries [140,145]. For example, Liraglutide countered CKD without metabolic syndrome through the downregulation of inflammatory gene expressions, including C3, CCL2, and TNFα [146]. Treatment with exendin-4 resulted in improved renal function, reflected by a decrease in creatinine clearance and urinary albumin excretion and suppressed glomerular hypertrophy and mesangial expansion [147]. Additionally, the decline in renal lipid accumulation of proinflammatory cytokines like IL-6, IL-1β, and TNF-α was also suggestive of the anti-inflammatory effect of Semaglutide [142].

Furthermore, GLP-1 agonists counter renal fibrosis in DKD. Particularly, the drug plays a significant role in preventing epithelial-mesenchymal transition (EMT), thereby blocking downstream cascades that cause ECM protein secretion [148,149]. Involving this mechanism, liraglutide inhibited pSmad3 and pERK1/2, downstream signalling molecules in the TGF-β1 pathway, and effectively reduced renal fibrosis, indicated by lowered fibronectin expression [144,146]. Furthermore, in unilateral ureteral obstruction injuries, such anti-fibrotic activity promoted a reduction in UUO-induced collagen deposition [146]. Fibrosis in CKD is decelerated as collagen deposition is weakened.

GLP-1 agonists are also found to slow CKD progression by other means. For example, Semaglutide is found to significantly ameliorate glucose homeostasis and insulin resistance, further improving renal outcomes in the disease course [142,150]. Moreover, Beinaglutide elevated the expression of both megalin and cubilin for reabsorption, enhancing the capacity for albumin absorption, and thus alleviating albuminuria [151]. Renal function restoration is reflected by reduced levels of kidney injury molecule-1 (KIM-1) in urine/renal tissue, improvement in glomerulosclerosis severity, enhanced podocyte filtration slit density, and a decline in albuminuria.

### 3.4. ACE Inhibitors and Angiotensin Receptor Blockers

ACEi and ARBs are both recommended as first-line treatments for hypertension [152]. ACEi, such as Benazepril, Captopril, and Ramipril, function by inhibiting the ACE enzyme, which converts angiotensin I to angiotensin II. This leads to vasodilation, natriuresis, and a decrease in sympathetic activity, ultimately reducing blood pressure. However, in patients with hypertension and heart failure undergoing chronic ACEi treatment, there is a potential for the “angiotensin escape” phenomenon. This is where angiotensin II can still be produced, negating the effects on the AT1 and AT2 receptors [153]. On the other hand, ARBs like Losartan, Candesartan, and Telmisartan work downstream in the renin-angiotensin system, preventing angiotensin II from binding to the AT1 receptor and thereby mitigating its adverse effects. The activation of the AT1 receptor, a primary activator of vascular NADPH oxidase, triggers harmful outcomes, including hypertension, oxidative stress, and inflammation [152,154].

In comprehensive meta-analyses evaluating the effect of ACEis and ARBs for treating CKD, several key findings regarding the efficacy of the individual drug and combination therapy emerged. ACEis and ARBs are found to have comparable clinical renal outcomes [155,156]. Both are found to be associated with slower CKD progression and lower risk of cardiovascular mortality [157,158,159]. Nevertheless, the benefit of combined utilization of ACEis and ARBs is debatable as some studies observed an enhanced renoprotective outcome reflected by a greater reduction in proteinuria, while other research suggested no significant difference in efficacy compared to that of individual drugs [155,160]. However, side effects, like coughing and bone degradation, may arise under treatment [161,162].

Recent studies have provided insightful advances in understanding the underlying mechanisms through which ACEi and ARBs may offer renal protection, mainly through anti-fibrotic and anti-apoptotic mechanisms. Angiotensin II promoted renal fibrosis progression when bound to angiotensin II receptor type 1, which triggered downstream profibrotic signalling pathways such as JAK-STAT and ERK [161,163]. Various ACEi and ARB target a wide range of pathways to exert their actions. For example, Captopril, one of the ACEi, hindered the phosphorylation and activation of JAK and STAT proteins in glomeruli, which suppressed apoptosis and fibrosis in diabetic nephropathies [162,164]. The anti-fibrotic activity was subsequently reflected by a decline in renal profibrotic agents, including CTGF and VEGF [162]. Gross and his team recognized a similar anti-fibrotic effect of Ramipril and Candesartan on non-hypertensive progressive renal fibrosis, indicated by reduced CTGF and TGFβ [165]. Additionally, restoration of renal function was observed through a significant reduction in kidney-related complications from diabetes, including albuminuria and glomerulosclerosis. Moreover, telmisartan downregulated type I and type IV collagen gene expression [166]. As a result, podocyte injury was ameliorated due to the halt in albumin cytoplasmic granule accumulation, and aggravation of albuminuria was prevented. The anti-fibrotic activity of ACEi is also exemplified by Imidapril, which inhibited DPP-4 and TGFβ signalling, subsequently upregulating the interaction among antifibrotic microRNAs (miR-29 and miR-let-7 family) in endothelial cells to weaken fibrosis [167].

### 3.5. Sodium-Glucose Co-Transporter 2

SGLT2 inhibitors, including canagliflozin, dapagliflozin, and empagliflozin, act as hypoglycaemic agents by inhibiting SGLT2 in the kidneys. This action reduces renal glucose reabsorption, leading to lower blood glucose levels without increasing insulin release [168,169]. Beyond their glucose-lowering effects, SGLT2 inhibitors have demonstrated beneficial effects on the kidneys by reducing the reabsorption of glucose in the kidneys. They alleviate the workload on the kidneys and reduce the production of ROS and inflammatory cytokines, which can cause oxidative stress-induced tubular impairment [170].

In light of the growing prevalence of diabetic nephropathies in T2DM patients, several meta-analyses have offered insights into the clinical utility of SGLT2 inhibitors. During the drug treatment course, an initial decrease in eGFR in the first weeks of the therapy is observed, followed by a stabilization of sustained kidney protection over time [171,172]. Compared with other drugs, it is effective in patients with very low eGFR (30 to 45 mL/min per 1.73 m^2^, ineligible for treatments) [173]. Moreover, the renoprotective outcome of the drug is boosted when combined with other interventions, notably RAS blockade for marked eGFR reduction. Clinically, the drug slowed down the progression of albuminuria and reduced the kidney-to-body weight ratio, indicating a reduced risk of kidney-related complications development and potential cardiovascular benefits, resulting in a lower overall mortality rate [174,175].

SGLT2 inhibitors counter ROS by acting as antioxidants. In DKD patients, SGLT2 is overactivated and subsequently causes increased glucose and sodium reabsorption. This results in hypertension and a heightened energy demand of renal cells, contributing to excess ROS generation [170]. By inhibiting the overreacting SGLT2, SGLT2 inhibitors lower the accumulation of ROS. Additionally, the drug modulates antioxidant and prooxidant activities. For instance, canagliflozin diminished high-glucose-induced ROS production through the PKC-NAD(P)H oxidase pathway by upregulating PKC and NOX4, thereby improving albuminuria and mesangial expansion [176]. Phlorizin is also found to reduce free radical species such as 3-nitrotyrosine (3-NT) and raise the antioxidant activities of catalase and glutathione peroxidase [177].

Moreover, SGLT2 inhibitors exert anti-inflammatory and anti-fibrotic activities to prevent DKD progression. This is exemplified by Dapagliflozin, which limited the elevated levels of HIF-1α and HIF-2α in renal proximal tubules, consequently suppressing inflammatory activities, such as macrophage infiltration, and fibrosis [178,179,180]. Such action also stopped PCT transitioning from fatty acid utilization to glycolysis and lipid accumulation, tackling the underlying cause of renal damage [180,181]. Hence, the drug improved diabetes-induced tubulointerstitial damage symptoms. Moreover, empagliflozin is found to work against proteinuric and non-proteinuric DKD through the modulation of ketone body level [181,182,183]. Within the damaged tubular cells, it suppressed the overreacting mTORC1 pathway, hence shifting the cells’ ATP production source from lipid to ketone. As endogenous blood ketone body levels, such as β-OHB, elevate under ketolysis, they prevent mTORC1-associated inflammation, which causes podocyte damage. As a result, renal function is improved.

**Table 1 antioxidants-13-00751-t001:** Therapeutic agents related to anti-inflammatory and antioxidant properties used for CKD management. Studies are organized according to the drug type to compare their targeted metabolic pathways or targets involving oxidative stress and the key findings summarise the anti-inflammatory and antioxidant effects of the drugs on different kidney disease models. The dosages and treatment duration are also outlined for each study.

Author	Therapeutic Agents	Name	Dose	Targeted Kidney Disease	Kidney Disease Model	Targeted Metabolic Targets/Pathway	Key Findings
Tamura et al. [105]	Statin	Pitavastatin, Rosuvastatin, and Pravastatin	0.005% (*w*/*w*) pravastatin, pitavastatin or rosuvastatin for 8 weeks	Diabetic nephropathy	db/db mice	Urinary8-OHdG levels	Pitavastatin, Rosuvastatin, and Pravastatin reduced oxidative stress by decreasing urinary 8-OHdG levels, with the added benefits of reduced albuminuria and glomerular hypertrophy.
Bruder-Nascimento et al. [102]		Atorvastatin	10 mg/kg/day for 2 weeks	DKD	db/db mice	ERK1/2, Akt/GSK3β, Nox4	Reduced oxidative stress, as evidenced by downregulation of ERK1/2, Nox4, upregulation of Akt/GSK3β, and reduction of ROS generation.
Zahr et al. [109]		Atorvastatin	10 mg/kg/day for 8 weeks	Sickle Cell Nephropathy	Townes humanized sickle-cell mice	NADPH oxidases, Cybb and Nox4	Reduced oxidative stress by a downregulation of NADPH oxidases, Cybb, and Nox4.
Singh et al. [103]		Atorvastatin	20 mg/kg/day for 8 weeks	Diabetic Nephropathy	Streptozotocin-treated Wistar rats fed with a cholesterol-supplemented diet	Histone deacetylase (HDAC)	Reduced diabetes-induced renal injury by the downregulation of HDAC activity and increased renal E-cadherin expression.
Zhang et al. [108]		Simvastatin	0, 0.1, 1, or 10 µM of Simvastatin and then exposed to Ang II for 24 h	CKD	Inflammation and oxidative stress induced by angiotensin II (Ang II) in human mesangial cells (HMCs)	COX-2, PKCs, NADPH oxidase, NF-κB p65, PPARγ, prostaglandin E2, TNF-α, IL-1β, and IL-6	Reduced inflammation and oxidative stress by downregulated COX 2, TNF-α, and NADPH oxidase activity.
Ota et al. [110]		Cerivastatin	0.1, 1.0 mg/kg/day for 12 weeks	Diabetic nephropathy	Spontaneously hypertensive rats (SHR) with streptozotocin-induced diabetes	MCP-1 and TGF-β	Reduced levels of inflammation marker MCP-1 and fibrosis marker TGF-β, with an added decrease in albuminuria
Yi et al. [127]	Metformin		2 mM/mL for 4 h for cell treatment, 0.4 mg/mL in drinking water for 14 days for mouse model	Renal tubulointerstitial fibrosis	HK2 cells, folic acid-induced mouse model of nephropathy	Smad3, ERK1/2, and P38, TGF-β1, MCP-1, F4/80	Reduced overexpression of TGF-β1 and improved renal injuries, inflammation, and fibrosis by targeting the TGF-β1 signalling pathway.
Lu et al. [128]			10 mM metformin for 0, 0.5, 1, 6, 12 h	Renal interstitial fibrosis	Primary cultured mouse renal fibroblasts	renal fibroblast collagen type I, CTGF, Smad3, AMPK	Reduced TGF-β1-induced collagen production through AMPK to limit Smad3-driven CTGF expression.
Yi et al. [124]			0.4 mg/mL metformin in drinking water for 21 days	Renal interstitial fibrosis	Adenine-induced renal injury mouse model	MCP-1, F4/80, ICAM1, type IV collagen and fibronectin, Smad3, ERK1/2, P38, AMPK	Decreased inflammatory and fibrotic indicators while inhibiting phosphorylation of molecules like Smad3, ERK1/2, and P38.
Thongnak et al. [131]			30 mg/kg/day for 8 weeks	renal dysfunction	Wistar rats with high-fat diet-induced insulin-resistant	AMPK, PPARα, SREBP1, FAS, Oat3	Combating renal lipotoxicity by activating AMPK/PPARα and inhibiting lipid metabolism pathways like SREBP1 and FAS.
Morales et al. [123]			150 mg/kg for 3 and 6 days	Gentamicin-induced acute renal failure	Gentamicin-treated rats	N-acetyl-β-D-glucosaminidase, cytochrome- c, and Mitochondrial NADH levels	Reduced gentamicin-induced nephrotoxicity by reducing N-acetyl-β-D-glucosaminidase activity against mitochondrial oxidative stress.
Ren et al. [133]			250 mg/kg/day for 8 weeks	Type 2 Diabetes Mellitus (T2DM)	High-fat diet and low-dose streptozotocin diabetic rat model and rat mesangial cells (RMCs) cultured with high glucose	AMPK, SIRT1, FoxO1	Reduced oxidative stress, increased autophagy, and reduced abnormal cell proliferation through the AMPK/SIRT1-FoxO1 pathway.
Zhang et al. [129]			70 mg/kg/day for 13 weeks	Type 2 diabetic nephropathy	High-fat diet and low-dose streptozotocin diabetic rat model	TGF-β1, CTGF	Ameliorated kidney dysfunctions and reduced levels of markers like TGF-β1 and CTGF, highlighting its renoprotective effects attributed to anti-inflammatory, anti-oxidative, and lipid-modulatory activities.
Li et al. [146]	GLP-1	Liraglutide	300 μg/kg every 12 h for 7 days	Renal fibrosis	Unilateral ureteral obstruction mouse model and TGF-β1-treated renal tubular epithelial cells	TGF-β1, Smad3, ERK1/2	Interrupts the epithelial-mesenchymal transition (EMT) process in renal fibrosis by downregulating TGF-β1 and its receptor and inhibiting molecules like pSmad3 and pERK1/2.
Dalbøge et al. [150]		Semaglutide	30 nmol/kg/day for 11 weeks	Type 2 diabetes	adeno-associated virus-mediated renin overexpression in the uninephrectomized diabetic db/db mouse	kidney injury molecule-1 (KIM-1)	Reduced hyperglycemia, hypertension, and albuminuria; enhanced kidney function by reducing levels of kidney injury molecule-1 (KIM-1).
Chen et al. [142]		Semaglutide	30 nmol/kg/day for 13 weeks	Obesity-related glomerulopathy	High-fat diet C57BL/6J mice model	Adenosine, NAD+, IL-6, IL-1β, and TNF-α	Ameliorated kidney injury by enhancing oxidative stress and inflammation-related kidney metabolites, specifically NAD+ and adenosine.
Yeung et al. [143]		Dulaglutide	0.6 mg/kg every 7 days for 4 weeks	Type 2 diabetes mellitus	High-fat diet C57BL/6J mice model	CDS1, PGPS, CLS, and TAZ	Protected against renal dysfunction by enhancing cardiolipin levels and upregulating cardiolipin synthesis genes, crucial for mitochondrial respiratory complexes.
Ougaard et al. [144]		Liraglutide	Accumulated 1 mg/kg within 4 days	Human CKD without metabolic syndrome (hyperglycaemia, dyslipidemia, and obesity),	Enalapril-treated mice	MAS1, CCL2, C3, C5aR1, CD3d, CD68, CXCL10, IL33, Itgam, JAK1, TNFα, Nox4 and VACM1	Improved eGFR and reduced renal fibrosis and inflammation by downregulating inflammatory genes like C3, CCL2, and TNFα and modulating angiogenesis, fibrosis, inflammation, and proliferation pathways.
Yin et al. [151]		Beinaglutide	1.5 pmol/kg/min for 12 weeks by osmotic pumps	Diabetic nephropathy	Streptozotocin-induced diabetes rats	PKC-β and PKA	Countered diabetic nephropathy by inhibiting PKC-β, boosting PKA activities, and affecting the expression of re-absorption proteins megalin and cubilin.
Zhang et al. [162]	ARBs	Captopril	10 mg/kg for 8 weeks	Diabetes mellitus	Lprdb/db (db/db) and Lprdb/+ (db/+) mice in C57BL/6-KS	CTGF and VEGF	Reduced profibrotic markers, CTGF, and VEGF, and potentially inhibiting the synthesis of ANG II.
Nishiyama et al. [166]		Telmisartan	10 mg/kg/day for 9, 22 weeks	Diabetic nephropathy	Otsuka Long–Evans Tokushima Fatty rats	type I collagen, type IV collagen	Blocked angiotensin II, leading to a reduction in systolic blood pressure, downregulation of type I and type IV collagen genes, and prevention of albuminuria increase.
Srivastava et al. [167]		Imidapril + AcSDKP	2.5 mg/kg/day (Imidapril) and 500 µg/kg/day (AcSDKP)	Fibrotic DKD	streptozotocin (STZ)-treated CD-1 mice	DPP-4, TGFβ, miR-29, miR-let-7	Inhibited markers of DPP-4 and TGFβ signalling and preserved miR-29 and miR-let-7 family interactions in endothelial cells.
Banes et al. [164]		Candesartan or Captopril	75–85 mg/kg/day for 2 week (Captopril) and 10 mg/kg/day for 2 weeks	Diabetes mellitus	Streptozotocin-induced diabetes rats	JAK2, STAT1, STAT3, and STAT5	Inhibited the activation of JAK and STAT proteins in rat glomeruli and suppressed phosphorylation of JAK2, STAT1, STAT3, and STAT5.
Gross et al. [165]		Ramipril or Candesartan	10 mg/kg/day for 3–6 weeks (Ramipril and Candesartan)	Non-hypertensive progressive renal fibrosis	COL4A3−/− mice	CTGF, TGFβ	Ramipril and Candesartan reduced renal fibrosis, and profibrotic markers, CTGF and TGFβ. Ramipril showed a stronger antifibrotic effect.
Cai et al. [180]	SGLT2 inhibitors	Dapagliflozin	1.5 mg/kg/day for 12 weeks	DKD	Streptozotocin-induced experimental mouse model and primarily cultured proximal tubule epithelial cells	HIF-1α	Limited the elevated levels of HIF-1α in the renal proximal tubule; improved symptoms of tubulointerstitial damage and suggested its potential as a therapeutic strategy for kidney tubule issues in DKD.
Tomita et al. [182]		Empagliflozin	30 mg/kg/day for 8 weeks	DKD	Damaged proximal tubules of high-fat diet-fed ApoE-knockout mice	Blood β-OHB level	Increased blood levels of the ketone body β-OHB, which might play a key role in its kidney-protective effects
Inada et al. [179]		Canagliflozin	40 mg/kg/day for 20–37 weeks	Diabetic nephropathy	Inducible cAMP early repressor transgenic (Tg) mice	HIF-1α and HIF-2α	Counteracted glomerulosclerosis and interstitial fibrosis by restoring abnormal HIF-1α and HIF-2α expressions.
Maki et al. [176]		Canagliflozin	0.01, 0.1, 1.0, 3.0 mg/kg/day for 8 weeks	Diabetic nephropathy	db/db mice	PKC, NOX4, fibronectin	Diminished high-glucose-induced ROS production and reduced TGF-β1 and fibronectin, indicating its reno-protective capabilities through possible inhibition of mesangial SGLT2.
Matthews et al. [184]		Dapagliflozin, Canagliflozin, Empagliflozin	25 mg/kg/day for 8 weeks	DKD	Kimba and Akimba mouse models	SGLT2	Slowed the progression of kidney disease, showcasing the therapeutic potential of these SGLT2 inhibitors for kidney health.
Lu et al. [185]		Empagliflozin	10 mg/kg/day for 12 weeks	DKD	db/db mice (C57BLKS/J-leprdb/leprdb)	Renal purine metabolism, Pyrimidine metabolism, Tryptophan metabolism, Nicotinate and nicotinamide metabolism, Glycine, serine, and threonine metabolism in serum	Modulated various renal metabolic pathways, indicating its potential in renoprotection by reducing oxidative stress and influencing specific amino acid transporters and enzymes in DKD.

## 4. Evaluating the Potential of Bioactive Supplements and Herbs as Antioxidative Agents in CKD Therapeutics

In the context of CKD, oxidative stress and inflammation are pivotal players in the advancement of renal dysfunction. Driven by metabolic abnormalities such as obesity, hypertension, and diabetes, these pathways inflict cellular damage and tissue fibrosis, worsening the intricate pathophysiological of CKD. While conventional pharmacotherapeutics such as statins, metformin, GLP-1, ACEi, ARBs, and SGLT2 inhibitors were initially developed to ameliorate metabolic syndromes, their renoprotective effects through antioxidative and anti-inflammatory mechanisms have been well discussed in the previous section. Consequently, this section shifts its focus to evaluating specific bioactive supplements and herbs, including Lactoferrin, *Boerhaavia diffusa*, *Amauroderma rugosum*, and *Ganoderma lucidum*, as potential adjuncts in the treatment of CKD with an emphasis on antioxidative effects. These compounds serve as examples of natural antioxidants that have shown potent anti-oxidative and anti-inflammatory properties in different disease models.

Lactoferrin is an iron-chelating glycoprotein found exogenously in bovine milk or endogenously in body secretions such as saliva [186]. Known to possess a variety of beneficial properties, exemplified by their anti-inflammatory, immunomodulatory, and anti-cancer properties, they have a wide spectrum of applications for improving human health [186,187]. It is often used in the pharmaceutical industry for disease therapy such as iron-deficiency anaemia and COVID-19 [188,189]. Recently, the utilization of lactoferrin as an additive in supplements that serve medical purposes, including infant formula and nutrient supplements, is also common in the food industry [187]. Nevertheless, as lactoferrin exhibits antioxidative and anti-inflammatory activities, potentially proving it as a potent renoprotective supplement by related mechanisms, its clinical applicability awaits further validation through human trials.

Boerhaavia diffusa (BD) has a long history of use in Ayurvedic medicine and is also utilized in other regions such as South America and Africa [190]. BD is rich in nutritional supplements and contains various beneficial compounds, including isoflavonoids, steroids, and phenolic compounds like Boervinones, quercetin, caffeoyltartaric acid, and terpenoids [191,192]. Experiments on in vitro and in vivo models have also suggested many therapeutic functions of BD such as anti-diabetic, anti-inflammatory, anti-cancer, and hepatoprotective and renoprotective properties [193]. These compounds extracted from BD function as direct antioxidants, attractants, and defence response chemicals [194]. They are crucial in combating damaging effects such as oxidative stress, ageing, and inflammation in the human body [194,195]. As a result, BD has been traditionally used in the treatment of various conditions, including asthma [196], gynaecological disorders [197], and urinary tract infections [198]. Given its traditional use and reported phytochemistry, BD is a promising potential candidate as a bioactive supplement for CKD.

*Amauroderma rugosum* (AR) is a type of polypore fungus that is used in traditional Chinese medicine. It belongs to the *Ganodermataceae* family [199]. In the traditional practices of the Temuan tribe in Peninsular Malaysia, AR is worn as a necklace to prevent fits, which are commonly known as epilepsy. It has also been associated with inflammation [200]. AR contains various phenolic compounds and triterpenes that have been shown to have antioxidant, anti-inflammatory, anti-apoptotic, anti-fibrotic, and immunomodulatory effects [200,201]. Neuroprotective, anti-cancer, anti-hyperlipidemic, anti-epileptic, and more recently, gastroprotective effects have been reported [199,202,203]. To the best knowledge of the authors, there are no reports of the renoprotective effects of AR hitherto, but the bioactivities of AR may potentially benefit patients with CKD, a condition characterized by oxidative stress, inflammation, fibrosis, and dysfunctional immunity [204,205]. It is also noteworthy that the antioxidative effect was shown to be even higher than *Ganoderma lucidum*, a more widely studied member of the same family, which could be attributed to more concentrated phenolic compounds in AR [206]. Therefore, AR is anticipated to have potent renoprotective effects and is considered a promising bioactive supplement that should be further investigated in clinical studies for its potential to manage and prevent the progression of CKD in humans. *Ganoderma lucidum* (GL) is a medicinal mushroom that originated from Chinese medicine and was believed to improve longevity. Containing bioactive compounds such as triterpenoids, LZ-8, polysaccharides, and polypeptides throughout different parts of the mushroom, GL commercial products were produced from the mushroom’s mycelia, spores, and fruit body [207,208]. Bioactive components in GL exhibit a large variety of health-promoting activities such as antioxidant, hypoglycaemic, anti-tumour, cardioprotective, and immunomodulatory effects in combination [209,210,211]. Nowadays, it is commonly used to treat a variety of highly prevalent diseases with high mortality rates such as cancer and cardiovascular and metabolic syndromes [210,212]. As GL consists of bioactive compounds with antioxidant and anti-inflammatory activity, its ability to ameliorate CKD has been widely investigated in recent years [212,213].

Despite their potential, the clinical translation of these bioactive supplements and herbs is limited by a lack of human studies, necessitating rigorous clinical trials for mechanistic explanation and therapeutic validation. A comprehensive overview of these bioactive supplements and herbs, summarizing current findings on targeting various diseases, related metabolic targets/pathways involving oxidant stress, and key findings of supplements and herbs has been provided in Table 2.

### 4.1. Lactoferrin

Lactoferrin, originally known as lactotransferrin, stands as a multifunctional iron-binding glycoprotein belonging to the transferrin protein family. It was first isolated from bovine milk in 1939 and later identified as the primary iron-binding protein in human milk [214,215]. Interestingly, it has also been found in different mammalian species including bovines, cows, goats, horses, and non-mammalian species like fish [216]. Lactoferrin is a multipurpose protein involved in a range of physiological and protective functions, such as regulating iron absorption in the gut and displaying antioxidant, anticancer, anti-inflammatory, and antimicrobial activities [217,218,219,220,221,222,223,224,225]. Aside from its abundance in mammalian milk and colostrum, lactoferrin is widely distributed in various bodily secretions like tears, saliva, and semen, as well as in neutrophil granules [226,227,228]. Structurally, the protein has a molecular weight of 80 kDa and is comprised of approximately 700 amino acids, with a high degree of homology observed between different species [222,229,230,231,232]. Given its broad physiological functions and high tolerance in humans, it has received FDA and European Food Safety Authority approval as a dietary supplement in food products [233].

The antioxidative effects of lactoferrin as well as its protective effects in various renal pathologies have been demonstrated in a number of studies. These studies showed that lactoferrin administration reduced albuminuria, blood urea, and creatinine levels in the renal disease model, which implied improved renal function [234,235,236,237]. Alleviation of renal damage by lactoferrin was further demonstrated by the reduction of kidney injury markers and histological findings [234,235,236]. Moreover, lactoferrin reduced inflammatory cytokine levels (e.g., IL-6), downregulated ERK1/2 and NF-κB pathways [236], and lowered CTGF levels [235,237], proving the anti-inflammatory and anti-fibrotic roles of lactoferrin. Notably, lactoferrin exerted antioxidative actions in its protection against renal diseases. It was shown to decrease oxidative stress and enhance antioxidative capacity by upregulating the Nrf2/HO-1 pathway [235,237]. As oxidative damages caused by ROS were prevented, kidney injury and failure could be alleviated, associated with the reduced morbidity and mortality of CKD patients [238]. Some studies suggested that other biological processes might also be involved in the protective effects of lactoferrin against renal diseases, including enhancing autophagy and reduction of necroptosis [234,237].

Despite these encouraging results, there are limitations to consider. The majority of studies [234,235,236,237] have predominantly used animal models or in vitro setups, which may not fully represent human physiology. Furthermore, the diverse range of metabolic markers targeted in these studies—AMPK, SREBP1, SREBP2, CSF2/CENPE, HIF-1α/VEGF, Akt/mTOR, IL-6, and now Nrf2/HO-1—suggests that our current understanding of lactoferrin’s antioxidant mechanisms remains incomplete. In summary, lactoferrin has demonstrated antioxidative and anti-inflammatory impacts through multiple biochemical markers and pathways across numerous studies. However, the need for more human clinical trials is evident to further validate these promising yet preliminary findings and to provide a more complete understanding of lactoferrin’s therapeutic potential, especially in CKD.

### 4.2. Boerhaavia diffusa

BD, also known as Punarnava, is a perennial creeping herb belonging to the *Nyctaginaceae* family. It is commonly found in various tropical and subtropical regions, including India, Brazil, Africa, Australia, and multiple countries in the Middle East. [239]. The six *Boerhaavia* species found in India include *B. diffusa*, *B. chinensis*, *B. erecta*, *B. repens*, *B. rependa*, and *B. rubicunda* [240]. Known by the Sanskrit name *Punarnava*, which translates to “one that rejuvenates the old body”, *B. diffusa* is a unique plant that dries up during the summer only to regenerate in the rainy season. It was named in honour of the 18th-century Dutch physician Hermann Boerhaave [240]. The plant is characterized by its sprawling branches and stout, fusiform roots. Its leaves are thick, fleshy, and hairy, while its small flowers range from pink to pinkish-red in colour [241]. Notably, the whole plant serves as the source for the drug Punarnava, which is recognized in the Indian Pharmacopoeia for its diuretic properties. BD holds significant ethnobotanical value and is traditionally used to treat different diseases, ranging from liver complaints and kidney disorders to cardiac conditions and general debility [242,243,244]. Its resilience and extensive therapeutic applications make it a subject of immense interest for both traditional medicine and scientific research.

BD has been extensively studied for its antioxidant and renoprotective properties in various kidney diseases. In CKD, it downregulated TGF-β, a marker commonly associated with renal fibrosis, signifying its antifibrotic and antioxidant potential [245]. In diabetic nephropathy, BD significantly impacts key antioxidant enzymes such as GPx, CAT, SOD, and GSH, restoring renal antioxidant status [246]. Remarkably, its effect on antioxidant enzymes rivals or surpasses that of metformin in targeting diabetes mellitus [247]. In the case of urolithiasis, BD protects against the oxidative stress and renal cell injury induced by calcium oxalate crystal formation [248]. Notably, Pareta et al. focused on hyperoxaluria and demonstrated that BD reduced oxalate excretion and malondialdehyde (MDA) and different antioxidant enzymes including SOD, CAT, GST, and GPx [249]. Oburai et al. further extended the research to treating chronic renal failure in dogs and found BD to be comparable to enalapril in reducing several markers of kidney function, such as serum creatinine, urea nitrogen, phosphorus, and urinary protein [250]. In particular, this herb was effective in normalizing potassium levels. Integrating the findings of these studies, it is signified that BD exerts its renoprotective effects by countering fibrosis and ROS accumulation in CKD conditions.

While these studies offer compelling insights into BD’s potential, they also echo limitations similar to those found in lactoferrin research. Many of these investigations [245,246,247,248,249,250] have been restricted to animal models, and the precise biochemical pathways underlying BD’s antioxidant mechanisms are still only partially understood. Hence, more human clinical trials are necessary for a detailed understanding of these mechanisms.

### 4.3. Amauroderma rugosum

AR, commonly referred to as “jiazhi” or “wuzhi” in traditional Chinese medicine, belongs to the *Ganodermataceae* family and is a unique basidiomycete mushroom. The fungus is found predominantly in tropical and subtropical zones, including regions such as China, South Pacific, South Atlantic, Indonesia, Taiwan, Equatorial Guinea, and Australia [199]. Characterized by its distinctive taupe-to-black cap, which is rugged and tomentose with a width ranging from 6–9 cm in diameter and 0.7–1.3 cm in thickness, AR has a hymenium that turns dark red upon being scratched, earning the name “Blood Lingzhi” in Chinese culture [251]. Despite being relatively underexplored in the scientific literature, existing literature reveals a complex phytochemical profile that includes sterols, flavonoids, fatty acids, phenolic compounds, and other bioactive elements [199,200,201,252,253,254]. These compounds have shown promise in preliminary research for their anti-proliferative, anti-inflammatory, and antioxidant properties, thereby suggesting potential applications for AR in the treatment of cancer and inflammatory diseases [199]. Given the limited scope of current research on AR, further studies are needed to validate these findings and explore additional therapeutic applications.

AR has demonstrated considerable promise in its antioxidant capabilities across various medical conditions, especially in age-related diseases [199]. In a study focused on gastric ulcers, Mai et al. observed that AR significantly reduced the size of gastric ulcers in ethanol and indomethacin-treated rats while lowering levels of inflammatory markers such as TNF-α, IL-6, and IL-1β [255]. Additionally, their study revealed that AR inhibited NF-κB P65 nuclear migration and downregulated NLRP3 gene expression. Li et al. focused on Parkinson’s disease and found that AR’s aqueous extract not only scavenged ROS in 6-OHDA-treated PC12 cells but also restored the downregulated Akt/mTOR and MEK/ERK signalling pathways [201]. Furthermore, Li et al. found upregulation of Nrf2 and HO-1 in Doxorubicin-induced cardiotoxicity models after AR extract administration [256]. Chan et al. presented its anti-inflammatory and antioxidant activity in LPS-treated murine macrophage RAW264.7 cells, through the effective scavenging of nitric oxide, ABTS, and DPPH radicals [200]. Similarly, another study conducted by Chan et al. confirmed AR’s ability to scavenge DPPH and ABTS radicals in the LPS-treated murine macrophage RAW264.7 cells [254]. Meanwhile, Shiu et al. demonstrated that AR reduced intracellular ROS levels and inhibited the release of oxidant-stress-related cytokines and chemokines like IL-1β and IL-8 in TNF-α- and IFN-γ-stimulated HaCaT keratinocytes [257]. The study also identified that AR can downregulate key signalling pathways related to oxidant stress, including NF-κB, MEK1/2, ERK1/2, and Akt/mTOR. Despite these promising findings, the existing studies often target specific metabolic pathways or markers without offering a comprehensive view of AR’s antioxidative mechanisms. This targeted focus limits our complete understanding of how AR functions as an antioxidant. Moreover, given that AR exhibits antioxidant effects, its action on the kidney has yet to be discussed in previous studies. As kidney damage is aggravated by ROS- and inflammation-related pathways, AR could exert a beneficial effect on renoprotection in CKD. Therefore, future research should adopt a holistic approach to better elucidate AR’s full antioxidative potential.

### 4.4. Ganoderma lucidum

GL, also known as “lingzhi” or “reishi”, a medicinal mushroom revered for its diverse bioactive components, has been a cornerstone in traditional Chinese medicine for over 2400 years. This fungus is highly esteemed for its capacity to promote health, longevity, and cognitive growth, primarily due to its potent immune-boosting attributes [258]. Among the bioactive compounds in GL, triterpenoids and polysaccharides are the most pharmacologically active, showing a range of medicinal effects like antimicrobial, antitumor, anti-inflammatory, and hypolipidemic activities [259,260,261,262,263,264,265,266,267,268,269]. Modern scientific research has further underscored its immunostimulant and antioxidant properties [270]. Polysaccharides derived from GL have been particularly noted for their antioxidant capabilities [271,272,273], and triterpenoids are recognized for their complex and highly oxidized chemical structures that contribute to the mushroom’s biological capacity [274,275]. Various purification methods, such as trichloroacetic acid (TCA) precipitation, enzymatic methods, and lead acetate precipitation have been used to purify these polysaccharides, each with its advantages and limitations [276,277,278,279,280]. Given the mushroom’s multifaceted bioactivity and historical relevance, GL continues to be an object of extensive research and a promising candidate for future nutraceutical and medicinal applications.

GL has garnered significant interest for its potential therapeutic applications in a range of diseases, particularly those affecting the liver and kidneys. This interest stems primarily from its antioxidant and anti-inflammatory properties. In terms of diabetic nephropathy, He et al. demonstrated that Ganoderma lucidum polysaccharides (GL-PS) significantly ameliorated metabolic abnormalities in streptozotocin-induced diabetic mice [281]. Specifically, GL-PS reduced serum creatinine and blood urea nitrogen levels while reducing oxidative stress regulation markers such as MDA and SOD. Pan et al. also reported a substantial increase in kidney antioxidant enzymes such as SOD, GSH-px, and CAT when treated with GL proteoglycan [282]. Similarly, Zhong et al. found that GL improved renal function in a renal ischemia-reperfusion injury model by diminishing ROS production and inhibiting stress-induced apoptosis [283]. Overall, these findings revealed GL’s potential to restore renal function, mainly through its antioxidative and anti-apoptotic actions.

For liver diseases, Wu et al. showed that GL extract reversed thioacetamide-induced liver fibrosis in mice, likely through enhancing collagenase activity [284]. Shi et al. found that GL-PS protected against chronic liver injury in D-galactosamine-treated mice by reducing oxidative stress markers such as MDA and liver damage markers such as AST and ALT [285]. Lin and Lin indicated that GLE effectively reduced carbon tetrachloride-induced liver fibrosis and oxidative stress, as evidenced by decreased MDA levels [286]. Lai et al. showed that GL reduced oxidative damage in proximal tubular epithelial cells, although their study focused more on inflammatory markers like IL-8 and sICAM-1 rather than on oxidative stress markers [287].

Despite the promising data on GL’s potential therapeutic benefits, gaps remain in the existing research, particularly with respect to understanding its antioxidant mechanisms. While the studies conducted by He et al. [281] and Pan et al. [282] highlight the potential antioxidant benefits of GL in treating kidney diseases, they did not identify the specific active components responsible for the antioxidant effects. On the other hand, Seto et al. focused on its impact on diabetes mellitus but did not explore its antioxidant pathways [288]. This lack of comprehensive analysis leaves important questions unanswered, particularly concerning the identification of the active components and the exact antioxidant mechanisms at play. These limitations suggest that existing research provided only a partial understanding of the specific antioxidant pathways and active components responsible for GL’s beneficial effects. To fully elucidate its mechanisms, particularly concerning its antioxidant effects, more comprehensive studies are needed.

**Table 2 antioxidants-13-00751-t002:** The potential of bioactive supplements and herbs in CKD antioxidative therapeutics. Studies are organized according to the supplement/herb type to compare their targeted metabolic pathways involving oxidative stress and the key finding summarises the anti-inflammatory and antioxidant effects of the supplement/herbs on different disease models. The dosages and treatment duration are also outlined for each study.

Author	Supplement/Herbs	Dose	Targeted Disease	Disease Model	Targeted Metabolic Targets	Key Findings
Aoyoma et al. [289]	Lactoferrin	0, 100, or 500 mg/kg/day for 17 weeks	Non-alcoholic steatohepatitis (NASH)	Connexin 32 dominant negative transgenic (Cx32ΔTg) rats fed with a high-fat diet (HFD)	TNF-α, IL-6, IL-18, IL-1β, TGF-β1, TIMP2, COL1a1, (NF)-κB	Reduced inflammation and fibrosis in a NASH rat model, potentially via NF-κB and TGF-β1 signalling pathways.
Alnahdi et al. [290]		0, 50 mg/kg/day for 30 days	Diabetic nephropathy and cardiomyopathy	Streptozotocintreated Wistar rats	AGEP, CTGF, TNFα, IL-6	Improved kidney and heart function in diabetic rats by suppressing CTGF expression and inflammatory cytokines TNF-α and IL-6.
Singh et al. [236]		22.07% LF diet for 8 weeks	Hypertensive stroke and nephropathy	High-fat-fed spontaneously hypertensive stroke-prone rats	Renin, osteopontin, MCP-1, IL-6	Decreased renal damage and delayed stroke onset in rats, with a significant reduction in glucose levels and downregulation of kidney damage markers.
Hsu et al. [237]		0, 100, 150, 200 μg/mL for 24 h (cells); 2, 4 mg/mouse twice a week for five weeks (mice)	Acute kidney injury	HK-2 cell, Folic acid-treated C57BL/6 mice	LTF, AMPK, Akt, mTOR, CTGF, PAI-1, and Collagen I	Obstructed renal fibrosis and reduced oxidative stress in kidney cells, notably inducing autophagy via the activation of AMPK.
Mohamed et al. [235]		0, 300 mg/kg/day for 7 days	Cyclophosphamide-induced nephrotoxicity	Cyclophosphamide-treated Sprague Dawley rats	Nrf2, HO-1, p-ERK1, p-ERK2, TNFα, IL-6, NF-κB, Wnt4, β-catenin, GSK-3β, klotho, caspase-3 and Bcl2	Lowered creatinine and blood urea nitrogen (BUN) levels in cyclophosphamide-treated rats, while modulating several kidneys protective signalling pathways such as downregulating ERK1/2 and NF-κB and enhancement in Nrf2/HO-1 signalling led to increased antioxidant capacity.
Liu et al. [234]		0, 10, 20, 30, 40 μg/mL for 6 h (cell treatment); 0, 2, 4 mg/mouse twice a week for 10 weeks (mice)	Particulate matter-induced nephrotoxicity	HK-2 cells and C57BL/6 mice exposed to particulate matter	CSF2, CENPE	Prevented particulate matter-induced kidney cell death by inhibiting necroptosis, while inducing autophagy through the CSF2/CENPE pathway.
Guo et al. [291]		0, 4 mg/kg/day for 40 days	Non-alcoholic fatty liver disease	Leptin-deficient (ob/ob) C57BL/6J mice	SREBP2, HIF-1α, VEGF, SOD1, JAK2, IL-6, Bax, Caspase3	Improved hepatosteatosis in ob/ob mice by regulating lipid and iron homeostasis, suppressing oxidative stress and inflammation and inducing hepatic autophagy.
Pareta et al. [249]	*Boerhaavia diffusa*	100, 200 mg/kg/day for 28 days	Hyperoxaluria	Ethylene glycol (EG)-treated Wistar albino rats	Urinary oxalate, serum creatinine, blood urea nitrogen (BUN), MDA, SOD, CAT, GST, GPx	Inhibited oxalate synthesizing enzymes, reducing urinary oxalate. The diuretic effect reduced oxalate saturation and prevented CaOx precipitation. Improved renal function markers (BUN, creatinine clearance). Mitigated oxidative stress markers and restored antioxidant enzyme activity. Inhibited crystal deposition in kidneys.
Sathees-h & Pari [246]		0, 200 mg/kg/day for 4 weeks	Diabetes mellitus	Alloxan treated rats	SOD, CAT, GPx, GST, GSH	Reduced lipid peroxidation markers (TBARS, hydroperoxides). Increased levels of Glutathione (GSH) and activity of antioxidant enzymes (SOD, CAT, GPx, GST). Contains compounds like alkaloids and sterols, responsible for antioxidant and antidiabetic effects.
Oburai et al. [250]		500 mg/dog/day for 90 days	Chronic renal failure (CRF)	Dogs suffering from CRF	Serum Creatinine, urea nitrogen, phosphorus, urinary protein, ALP, GGT	Improved symptoms and fewer deaths compared to the enalapril group. Reduced systolic and diastolic blood pressure. Normalized serum levels of urea nitrogen, creatinine, sodium, phosphorus, and potassium. Reduced markers of renal damage (ALP, GGT, urinary protein levels).
Sadayan et al. [245]		400 mg/kg/day for 14 days	CKD	Adenine-treated Wistar albino rat	Urea, serum creatinine, TGF-β	Reduced harmful serum markers like creatinine, urea, and glucose. Improved haematological parameters including red blood cells, epithelial cells, and urinary parameters including albumin levels. Downregulated TGF-β expression, indicating antifibrotic potential.
Singh et al. [247]		500 mg/kg/day for 30 days	Diabetic nephropathy	Alloxan-treated Wistar albino rat	Serum urea, creatinine, GPx, Catalase, SOD, GSH	Reversed loss of body weight and renal protein content. Reduced diabetic symptoms like increased water and food intake. Significant hypoglycaemic effect and increased insulin levels. Improved ionic homeostasis and enzyme activity (Na+–K+ ATPase).
Pareta et al. [248]		100, 200 mg/kg/day for 28 days	Urolithiasis	Ethylene glycol-treated Wistar rats	Calcium oxalate (CaOx) crystallization	Fewer and smaller CaOx crystals in urine post-treatment. Beneficial impact on crystal morphology. Prevented elevated oxalate and calcium levels associated with kidney damage. Protected against oxidative stress and renal cell injury induced by crystal formation.
Mai et al. [255]	*Amauroderma rugosum*	50, 100, 200 mg/kg for 7 days	Gastric ulcer	Ethanol and indomethacin treated Sprague Dawley rat	TNF-α, IL-6, IL-1β, PGE2, NLRP3, NF-κB	Reduced ethanol-induced gastric injuries and reversed elevated serum NO levels, implicating antioxidant properties for ulcer healing. Increased serum PGE2 levels, further aiding in ulcer healing. Lowered inflammatory cytokines such as TNF-α, IL-6, and IL-1β, while suppressing NLRP3 and NF-κB signalling pathways, indicating comprehensive anti-inflammatory mechanisms
Chan et al. [200]		0.01–100 μg/mL for 24 h	Inflammatory disorders	LPS-treated murine macrophage RAW264.7 cells	DPPH, ABTS, Nitric Oxide	A. rugosum mycelia are nutrient-rich, antioxidant, and anti-inflammatory properties. Ethyl linoleate and ergosterol contribute to anti-inflammatory effects
Chan et al. [254]		0.1–100 μg/mL for 24 h	Chronic inflammation	LPS-treated murine macrophage RAW264.7 cells	TNF-α, IL-10, NF-κB, DPPH, ABTS	Both wild and domesticated versions of Amauroderma rugosum downregulated TNF-α proinflammatory cytokines and upregulated IL-10 anti-inflammatory cytokines, but did not affect NF-κB translocation
Seto et al. [288]	*Ganoderma lucidum*	0.003, 0.03 and 0.3 g/kg/day for 4 weeks	Type 2 diabetes mellitus	db/db mice	PEPCK	Reduced plasma glucose levels without affecting insulin, suggesting that hypoglycaemic effects are not insulin dependent. Suppressed hepatic PEPCK gene expression, contributing to glucose regulation. It did not impact HMG CoA reductase, dismissing its role in cholesterol synthesis. Reduced body weight in obese/diabetic mice, likely due to abdominal fat reduction.
Pan et al. [282]		75, 250, 450 mg/kg/day for 8 weeks	Diabetic nephropathy	C57BL/6J db/db mice	serum creatinine, urea nitrogen, urea acid, albuminuria, SOD, GSH-px, CAT	Lowered blood glucose levels and protected pancreatic β-cells, implying potential in managing diabetes. Exhibited hypotriglyceridaemia and hypocholesterolaemia effects, potentially preventing diabetic complications. Confirmed renal-protective roles against diabetic nephropathy by altering key biochemical markers and suppressing oxidative stress.
Zhong et al. [283]		100 mg/kg/day for 7 day	Renal ischemia-reperfusion injury	C57BL/6J mice with renal ischemia-reperfusion injury and tunicamycin-treated NRK-52E cells	Bax, Bcl-2, caspase-3, GRP78, CHOP, caspase-12, JNK	Mitigated renal ischemia-reperfusion injury by balancing oxidative stress markers and inhibiting ROS production. Alleviated mitochondrial and endoplasmic reticulum stress-induced apoptosis, suggesting broad renal protective activities.
Lai et al. [287]		4, 8, 16, 32, 64 μg/mL for 24 h	Tubulointerstitial injury	Albumin-treated human proximal tubular epithelial cells	IL-8, sICAM-1	Demonstrated immunomodulatory effects and protected against cytotoxicity and DNA damage in renal cells. Indicated anti-inflammatory properties by reducing specific cytokine release.
He et al. [281]		125, 250 mg/kg/day for 8 weeks	Diabetic nephropathy	Streptozotocin-treated C57bl/6J mice	MDA, SOD, TGF-β	Reduced urinary albumin excretion and improved key renal function markers, supporting a protective role in diabetic nephropathy. Also ameliorated hyperglycaemia and oxidative stress
Wu et al. [284]		0.5, 1.0 g/kg/day for 4 weeks	Liver fibrosis	Thioacetamide-treated BABL/c mice	MMP-13, TIMP-1, collagen a1	Reversed liver fibrosis and modulated extracellular matrix degradation in mice. Improved body and liver weight, suggesting hepatic recovery. Altered the MMP-13/TIMP-1 ratio, implying a role in extracellular matrix remodelling.
Shi et al. [285]		60, 120, and 180 mg/kg/day for 2 weeks	Chronic liver injury	D-galactosamine-treated Kunming mice	AST, ALT, MDA, SOD, GSH	Reduced liver damage markers AST and ALT, indicating hepatoprotective effects. Demonstrated antioxidative capacities by maintaining liver enzyme activities and reducing MDA oxidative stress marker
Lin & Lin [286]		600, 1600 mg/kg/day for 8 weeks	Liver fibrosis	Carbon tetrachloride-treated Wistar rats	MAT1A, MAT2A, TGF-β1, MDA	Prevented liver injury by reducing plasma ALT and AST levels, and reduced MDA oxidative stress marker. Modified hepatic enzyme expression, indicating hepatoprotective effects. Improved liver protein and albumin levels, suggesting reduced liver inflammation and fibrosis.

## 5. Conclusions

This review evaluated existing evidence on the potential of antioxidative therapies in treating CKD by addressing inflammation and oxidative stress. Clinically used drugs like statins, metformin, GLP-1 agonists, ACEi, ARBs, and SGLT2 inhibitors show promise due to their renoprotective effects linked to anti-inflammatory and antioxidant activities. In addition, natural supplements and herbal medicines, such as lactoferrin, *B. diffusa*, *A. rugosum*, and *G. lucidum*, have been shown to exhibit antioxidative properties in disease models, both in vitro and in vivo. Animal disease models include HFD-Wistar rats, C57BL/6 mice, Sprague Dawley rats, dogs, db/db mice, and human proximal tubular epithelial cell tissue culture. These studies have demonstrated the ability of these compounds to modulate cytokines, oxidative stress markers, and antioxidant enzymes, showcasing their potential therapeutic benefits. While preliminary data are encouraging, there remain gaps in fully elucidating the underlying antioxidant mechanisms of these therapies. Further research using comprehensive approaches is needed to identify specific active components and their modes of action. Comprehensive human clinical trials are also required to validate promising findings from animal and cell studies. Quantification methods such as measuring the biomarkers of oxidative stress/damage (e.g., markers of lipid peroxidation, protein oxidation, DNA damage), antioxidants levels (e.g., glutathione, superoxide dismutase), and inflammatory cytokines using techniques like enzyme-linked immunosorbent assays (ELISAs) and multiplex immunoassays could help characterize the specific antioxidant pathways modulated and determine the magnitude of response to these therapies. If modulating oxidative stress and inflammation are validated as an effective strategy for managing CKD progression, it may expand therapeutic options. Integrated treatment approaches combining pharmacotherapies, natural products, lifestyle modifications, and anti-oxidative supplementation warrant investigation. Ongoing research in this area has the potential to establish modifiable oxidative balance as a modifiable risk factor, improving CKD outcomes through multi-target therapies aimed at restoring redox homeostasis. Further exploration of anti-oxidative strategies holds promise for developing practical applications supporting kidney health.

## Figures and Tables

**Figure 1 antioxidants-13-00751-f001:**
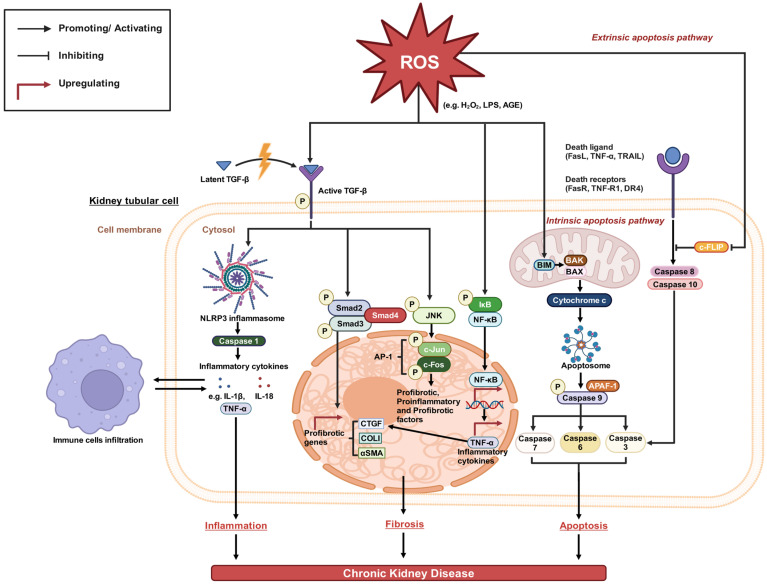
Oxidative stress-mediated pathways in the pathogenesis of CKD [27,35,36]. ROS from various sources, including H_2_O_2_, LPS, and AGE, can activate latent TGF-β through oxidative post-translational modifications. Activated TGF-β promotes fibrosis via canonical Smad signalling and non-canonical JNK and NF-κB pathways, increasing profibrotic gene expression. This enhances myofibroblast development and pathological fibrosis. ROS may also trigger the intrinsic and extrinsic apoptosis pathways in renal cells. Collectively, these oxidative stress mechanisms may underlie CKD pathogenesis.

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
