# Peer review of "Potentials of Natural Antioxidants in Reducing Inflammation and Oxidative Stress in Chronic Kidney Disease"

_antioxidants, 2024, doi:10.3390/antiox13060751_

Round 1

Reviewer 1 Report

Comments follow throughout the attached document.

Comments follow throughout the attached document.

Reviewer 2 Report

Chronic kidney disease (CKD) is a major health burden worldwide. Modern lifestyle, disruptive food and sleep habits are suspected to be contributing factors to CKD. Not surprisingly, diabetes emerged as the leading cause of CKD. In this review article, authors explained the causative factors of CKD and associated signaling pathways in great detail. Authors further discussed the different treatment strategies explaining the use of metabolic therapeutics like statins, anti-diabetic drugs like metformin, ACE inhibitors, etc. These drugs are although effective in managing CKD, the use of natural compounds and supplements gained lots of attention recently and authors reviewed their potential in treating CKD. Overall, this review article is well written and highly significant.

No specific comments, however, addition of recent references (papers published in 2024) will be helpful.

Reviewer 3 Report

The review manuscript aimed to address the effects of antioxidants on chronic kidney disease (CKD) treatments. In general, the authors have presented the potential treatment pathways on the CKD from the relative references. And, the manuscript has provided the possible treatment strategies on CKD and has confirmed to the aims of the study. However, the major concerns are focused in the contents after Line 566, i.e. Evaluating the Potential of Bioactive Supplements and Herbs as Antioxidative Agents in Chronic Kidney Disease (CKD) Therapeutics. The paragraph should be updated furtherly to confirm the aims of the study, owing to be lacking the relative in vivo or clinical studies. 

1.     There are countless antioxidants existed in the natures. The authors are requested to examine the appropriateness of the manuscript title shown in the study.

2.     The reason why the authors take the phytochemical examples of Boerhaavia diffusa, Amauroderma rugosum, and Ganoderma lucidum being applied in the review article.

3.     In the Figure 1, there are several references should be cited in the figure legend and should be given the discussion.

Round 2

Reviewer 1 Report

The indications are mentioned in the document sent to the authors.

The indications are mentioned in the document sent to the authors.

Round 3

Reviewer 1 Report

None.

None.